# Dynamical Downscaling of Surface Air Temperature and Wind Field Variabilities over the Southeastern Levantine Basin, Mediterranean Sea

**Mohamed ElBessa** [1,2], **Saad Mesbah Abdelrahman** [3], **Kareem Tonbol** [3] **and Mohamed Shaltout** [2,*]

1   Maritime Postgraduate Studies Institute (MPI), Arab Academy for Science, Technology and Maritime Transport (AASTMT), Abu-Qir, Alexandria 1029, Egypt; M.elbessa5000@student.aast.edu
2   Oceanography Department, Faculty of Science, Alexandria University, Alexandria 21526, Egypt;
3   College of Maritime Transport and Technology, Arab Academy for Science, Technology and Maritime Transport, Alexandria 1209, Egypt; saad.mesbah@aast.edu (S.M.A.); ktonbol@aast.edu (K.T.)
*   Correspondence: mohamed.shaltot@alexu.edu.eg ; Tel.: +20-1005255939

**Abstract:** The characteristics of near surface air temperature and wind field over the Southeastern Levantine (SEL) sub-basin during the period 1979–2018 were simulated. The simulation was carried out using a dynamical downscaling approach, which requires running a regional climate model system (RegCM-SVN6994) on the study domain, using lower-resolution climate data (i.e., the fifth generation of ECMWF atmospheric reanalysis of the global climate ERA5 datasets) as boundary conditions. The quality of the RegCM-SVN simulation was first verified by comparing its simulations with ERA5 for the studied region from 1979 to 2018, and then with the available five WMO weather stations from 2007 to 2018. The dynamical downscaling results proved that RegCM-SVN in its current configuration successfully simulated the observed surface air temperature and wind field. Moreover, RegCM-SVN was proved to provide similar or even better accuracy (during extreme events) than ERA5 in simulating both surface air temperature and wind speed. The simulated annual mean T2m by RegCM-SVN (from 1979 to 2018) was 20.9 °C, with a positive warming trend of 0.44 °C/decade over the study area. Moreover, the annual mean wind speed by RegCM-SVN was 4.17 m/s, demonstrating an annual negative trend of wind speed over 92% of the study area. Surface air temperatures over SEL mostly occurred within the range of 4–31 °C; however, surface wind speed rarely exceeded 10 m/s. During the study period, the seasonal features of T2m showed a general warming trend along the four seasons and showed a wind speed decreasing trend during spring and summer. The results of the RegCM-SVN simulation constitute useful information that could be utilized to fully describe the study area in terms of other atmospheric parameters.

**Keywords:** dynamical downscaling; Mediterranean Sea; Levantine; RegCM-SVN; surface air temperature; surface wind; climate; trend analysis

## 1. Introduction

The Southeastern Levantine basin (SEL) extends from 29° N to 33° N and from 27° E to 37° E, as shown in Figure 1. This basin represents an important environmental resource for natural gas exploration, farming, and fisheries. The study area lies in a transition zone between the hot, arid climate of North Africa and the cold, humid climate of Central Europe. Thus, SEL is influenced by interactions between mid-latitude, tropical, and sub-tropical processes [1–3]. The SEL climatic conditions display seasonal variation.

During the summertime, the Indian summer monsoon (ISM) and the Azores High are the main factors that affect SEL [4–6]. The ISM (which is associated with a thermal low) expands northwest towards the Mediterranean region through the formation of the Persian trough [7]. At the same time, the Azores High (which is associated with a sub-tropical maritime climate) expands eastwards. Based on ISM and the Azores High, a pressure dipole

(sharp east–west pressure gradient) dominates over the Eastern Mediterranean, which leads to persistent northerly winds (Etesians) [8]. On one hand, in the periods when the ISM dominates over SEL, the temperature increases greatly with very dry conditions over the SEL. On the other hand, when the Azores High dominates over SEL, the temperature and dry conditions are moderate.

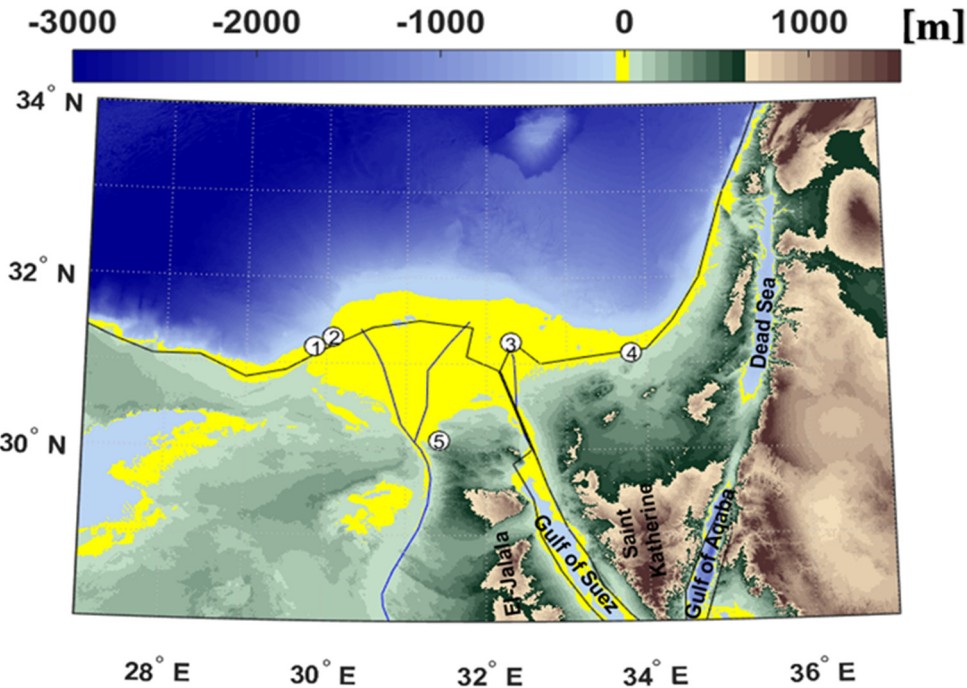

**Figure 1.** Bathymetric chart of the study area with the five considered weather observation stations: (1) Ras El Tin, (2) Abu Qir, (3) Port Said, (4) El Arish, and (5) Cairo airport, as well as the Gulfs of Suez and Aqaba together with the mountains El Jalala and Saint Katherine.

During the wintertime, SEL is highly affected by the Siberian High and the Azores High, together with primary and secondary low-pressure systems. The Siberian High system plays an important role in the climate/weather variation over the Eastern Mediterranean Sea and its surrounding landmass, as stated by De Vries et al. [9]. The Siberian High system, which is responsible for polar outbreaks over the Mediterranean area [10], (1) tends to produce favorable conditions for the rainy season to develop that paired with the Mediterranean low-pressure system, as stated by Nastos and Zerefos [11]; and (2) generates easterly storms over the Levantine basin [12]. In general, the southward movement of the Siberian High and Azores High (winter subtropical high-pressure cells) creates favorable conditions for precipitation and for the development of extratropical cyclones throughout the Mediterranean area [13]. Furthermore, Zerefos et al. [14] found that, as the Azores High moves southward, storm systems from the Atlantic tend to penetrate over the Mediterranean. In addition to that, the extreme weather events over SEL during the winter season are highly associated with the Mediterranean cyclones and rarely develop under different circumstances [15].

In general, the Nile Delta coast (as a part of SEL) increases the SEL socioeconomic importance, thanks to its high population densities, good agricultural land, and high tourism activities. Moreover, the study area is considered an extreme hotspot of climate vulnerability [16–18]. Understanding the long-term variabilities of surface air temperature and the wind is essential for the adaptation and mitigation plans over such hotspot marine areas. Recently released global atmospheric analyses represent a step-change tool for climate investigations; however, their resolution is still inadequate to describe regional/local climates and a downscaling procedure is needed to provide more reliable data to policy makers. Unfortunately, limited dynamical downscaling efforts have been exerted over

Egypt [19,20]. These few studies have not focused on the meteorological physical properties over SEL, highlighting the need for further efforts to improve our understanding. Thus, building an accurate dynamical downscaling system to simulate the climate of the study area would be extremely beneficial for a better understanding of its climatology.

The current paper is presented in four sections. The data and methods of analysis used (dynamical downscaling, statistical analysis, and validation) are covered in Section 2. In Section 3, the results are presented, while the summary and conclusion are included in Section 4.

In the current study, regional climate model, Apache Subversion (RegCM-SVN), is used to simulate near surface air temperature (T2m) and surface wind field over SEL during the period from 1979 to 2018 to test the dynamical downscaling ability to describe the climate of the study area with a similar or even better accuracy than ERA5. The dynamical downscaling results were validated, first against ERA5 databases and then with the corresponding atmospheric observations. After the validation, the results were statistically analyzed and used to examine the spatiotemporal and trend variabilities of T2m and surface wind field. As such, understanding the variabilities of T2m and wind speed ($UV_{10}$) in this region is the main goal of this study for planning adaptation measures in different life sectors (e.g., agricultural and energy sectors), together with finding suitable regional climate policies to cope with climatic change issues.

## 2. Data and Methods of Analysis

### 2.1. Data Used

2.1.1. Data Used to Force RegCM-SVN

1. Data of air temperature, geopotential height, relative humidity, and zonal/meridional wind components were obtained from ERA5 hourly data on 38 different pressure levels from 1979 to 2018 (https://cds.climate.copernicus.eu/cdsapp#!/dataset/reanalysis-era5-single-levels?tab=form [accessed on 28 September 2020]) [21] with a grid spacing of 0.25° × 0.25°. Replacing the old version of ERA-Interim, ERA5 provides data on a denser spatial and temporal grid together with a significant improvement in core dynamics and model physics [22]. ERA5 presents a long-term record of world climate and weather by assimilating observational data (from ground sensors and satellites) using the integrated forecasting system (IFS). It is an improvement over earlier re-analyses to be used for climate and meteorological scientific analyses [23].
2. ERA5 sea surface temperature (SST) is also obtained using the ERA5 reanalysis database during the years of 1979–2018 using ERA5 reanalysis on a single level database during the years of 1979–2018 [24].
3. Static surface dataset is freely available via (http://climadods.ictp.it/Data/RegCM_Data/SURFACE/ [accessed on 15 August 2020]) [25] and used to describe RegCM-SVN surface boundary conditions:

    - GTOPO_DEM_30s, Digital Terrain Model Elevation with 30 arc seconds ($\frac{1}{120}° \times \frac{1}{120}°$) spatial grid point.
    - GLCC_BATS_30, Global Land Cover Characteristics with a spatial grid point of 30 arc seconds.
    - GLZB_SOIL_30s, STATSGO/FAO soil texture with a spatial grid point of 30 arc seconds.
    - ETOPO_BTM_30s, lake bathymetric datasets with a spatial grid point of 30 arc seconds.

2.1.2. ERA5 Hourly Data on Single Levels from 1979 to 2018

Data of T2m and surface wind were obtained from ERA5 on an hourly basis from 1979 to 2018 with a spatial resolution of 0.25° × 0.25°. These data were used to validate the RegCM-SVN model results to gain confidence and a full image of the variabilities of T2m and $UV_{10}$ over the study area. This was done according to C3S [23] and Hersbach et al. [22],

as mentioned in Section 2.1.1; therefore, ERA5 can be used efficiently to describe the current dynamics of the atmospheric parameters.

### 2.1.3. WMO Observed Data from 2007 to 2018

Observed T2m and $UV_{10}$ data at five automated weather observing systems (AWOSs) along the study coastal area were used: at Ras El-Tin, Abu Qir, Port Said, and El Arish, as shown in Figure 1 and Table 1. The AWOSs were installed and maintained according to WMO regulations; air temperature and wind field records were calibrated to WMO standard heights.

**Table 1.** Positions and elevations of meteorological stations. The period of recorded data for all stations is from 1 January 2007 until the end of December 2018. The identification numbers (INs) 1 to 5 are also included in relation to Figure 1.

| Station | | International Station Number | Geographic Position | | Number of Observation | Height above Sea Level (m) |
|---|---|---|---|---|---|---|
| Names | IN | | Latitude | Longitude | | |
| Ras El Tin | 1 | 62,317 | 31°11′50″ | 29°51′49″ | 105,192 | 22 |
| Abu Qir | 2 | 62,320 | 31°19′55″ | 30°5′6″ | 105,096 | 27 |
| Port Said | 3 | 62,334 | 31°15′19″ | 32°18′17″ | 105,192 | 20 |
| El Arish | 4 | 62,331 | 31°08′54″ | 33°49′27″ | 105,192 | 15 |
| Cairo airport | 5 | 62,366 | 30°06′ 41″ | 31°24′ 50″ | 104,068 | 74 |

The strength of this dataset is that it provides continuous hourly observations over five sites (four of them located along the SEL coast; however, the fifth site is located inland) from 2007 to 2018 that helps for comprehensive weather analysis over both coastal and inland sites. The only gap was in Abu Qir data station with only 96 missing hours (from 18 November 2018 at 10:00 to 22 November 2018 at 09:00).

### 2.2. Method of Analyses

#### 2.2.1. RegCM-SVN Model

The RegCM-SVN is a regional climate model used to dynamically downscale the atmospheric features at a finer resolution and it is designed to be used for long-term regional climate simulations [26]. RegCM-SVN is a compressible and hydrostatic model that runs on an Arakawa B-grid with sigma-p vertical coordinates. The RegCM-SVN dynamical component is based on the hydrostatic version of the fifth generation Mesoscale Model (MM5; Grell et al. [27]), with enhancements to the coupling with a complex land surface model.

In the present study, RegCM-SVN is used to downscale ERA5 databases with a grid spacing of 0.25° × 0.25°. However, its earlier version (RegCM4) was used to downscale ERA-Interim with a spatial resolution of 0.75° × 0.75°. Therefore, it is expected that the current RegCM-SVN version may improve the regional atmospheric simulation compared with the earlier version. The current RegCM-SVN version is designed to use only the following atmospheric parameters: air temperature, geopotential height, relative humidity, and zonal/meridional wind components for dynamical downscaling simulation.

The model configuration for RegCM-SVN is as follows: (1) the radiative transfer by using the radiative transfer scheme of the global model CCM3 [28]; (2) the planetary boundary layer (PBL) following Holtslag scheme [29]; (3) land surface processes by applying the biosphere-atmosphere transfer scheme (BATs; Dickinson et al. [30]); (4) cumulus convection using Grell scheme [31]; (5) large-scale precipitation using the SUBEX scheme [32]; and (6) ocean fluxes following the Zeng scheme [33].

This study is designed to simulate the surface air temperature and wind field over a domain extended from 26° to 38° E and from 28° to 34° N, comprising an extended SEL area with a buffer zone. This simulation downscaled ERA5 to a finer spatial resolution of 10 km (0.09°) over 40 years from 1979 to 2018. One year of a spin-up (1978) was used to reach a state of statistical equilibrium at the beginning of 1979. The input file of 1978 is just a climatic average value.

2.2.2. RegCM-SVN Verification Analyses

Verification analyses for RegCM-SVN simulations were done using two approaches. First, bias and correlation analyses were performed between RegCM-SVN simulations and ERA5 to examine and verify the quality of RegCM-SVN simulation compared with ERA5 for T2m and wind speed components at a height of 10 m (zonal wind speed $U_{10}$, meridional wind speed $V_{10}$). In the second approach, a comparison was also performed between the simulated results, ERA5, and the observed data available from 2007 to 2018 (over the five mentioned observed stations). The correlation coefficient is calculated after and before removing the seasonal cycle to examine whether or not the correlation between observed data from one side and RegCM-SVN/ERA5 data sets for the other side depended on a seasonal cycle. Moreover, the root mean square error (RMSE) was calculated (Equation (1) to examine which of RegCM-SVN or ERA5 describe the observed data more accurately, especially during extreme weather conditions.

$$\text{RMSE} = \sqrt{\frac{1}{nj} \sum_{i=1}^{nj} \left\{ WMO_j - \text{model} \left( \text{lat}_j, lon_j \right) \right\}^2} \tag{1}$$

where

$nj$: amount of data measured, $j$: the identification numbers of WMO station ($j$ = 1, 2, ... 5); model: RegCM-SVN or ERA5; lat$_j$: latitude of the nearest grid-point containing the WMO $j$-station location; and $lon_j$: longitude of the nearest grid-point containing the WMO $j$-station location.

Extreme T2m weather events are identified when the observed values are greater than the annual T2m mean +2* annual T2m standard deviation or lower than the annual T2m mean −2* annual T2m standard deviation. In the same context, extreme UV10 weather events are identified when the observed values are greater than the annual UV10 mean +2* annual UV10 standard deviation. RegCM-SVN and ERA5 grid points that contained observation stations were the only considered grids for the comparison process (using the nearest neighbor algorithm).

Furthermore, analyses of skewness coefficient (a measure of the asymmetry of the probability distribution, where negative value denotes to longer left tails and positive value denotes to longer right tails) were performed over the five mentioned observed stations between the RegCM-SVN/ERA5 results and the observed data available to examine which of RegCM-SVN or ERA5 follows the probability distribution of the observed data.

2.2.3. Spatial and Temporal Distribution of Annual T2m, $U_{10}$, and $V_{10}$ over SEL

RegCM-SVN daily simulations were used to characterize the spatiotemporal features of T2m, wind speed, and wind direction over SEL for the study period 1979–2018, focusing only on inter-annual/annual variability. In addition, the current daily simulations were also used to calculate annual linear trends using ordinary least squares estimation. All linear trends were subjected to a t-test to confirm their significance.

The annual simulated mean over the study period was computed as the mean of daily simulated means along a specific year for the period from 1979 to 2018. In addition, the maximum (minimum) annual mean T2m represents the annual average values of the warmest (coldest) year. Moreover, the maximum (minimum) annual mean $UV_{10}$ represents the annual average values of the windiest (calmest) year.

In addition to that, the annual mean (from 1979 to 2018) over SEL was computed by spatially averaging the annual simulated mean for each grid point over SEL. Moreover, the annual trend (from 1979 to 2018) over SEL was also computed by spatially averaging the annual simulated trend for each grid point over SEL.

2.2.4. Seasonal Characteristics of Surface Air Temperature and Wind Speed

Daily surface air temperature and surface wind speed simulated by RegCM-SVN were used to study their seasonal characteristics, which were computed by spatially averaging

each seasonal mean (from 1979 to 2018) for each grid point over SEL. The winter season extends from December to February, spring extends from March to May, summer extends from June to August, while autumn extends from September to November.

As 95% of the values will be within 2 standard deviations of the mean, the years in which the mean temperature of a given season is greater than 2 standard deviations above the climatic mean of the same season are the warmest. Similarly, years in which the mean temperature of a given season is less than two standard deviations below the climatic mean are the coldest.

### 2.2.5. Variability of SEL Surface Air Temperature and Surface Wind

(a)    Seasonality analysis (Fourier analysis)

Variability (seasonality and relative phase) of T2m over SEL was studied using Fourier analysis based on the daily RegCM-SVN simulations for T2m. The amplitude and phase angle of a one-year cycle to each grid point were calculated in a similar way to Shaltout [34]. According to Parsons et al. [35], the inherent compromise that exists between time resolution and frequency is the major source that limits Fourier transformation results. Moreover, a finite time series always has a Fourier transform [36]. In the current study, a long period (1979–2018) of a finite daily time series was used for better detection of the T2m annual cycle. Furthermore, the relative phase is used to calculate the delay of the calculated annual cycle based on RegCM-SVN simulation with respect to the ideal annual cycle. The ideal annual cycle is a sinusoidal cycle reaching its maximum (minimum) values at day 91.3 (273.93) of calendar year. Wind speed and direction were not subjected to Fourier analyses, owing to the difficulty in identifying a significant periodical cycle for wind speed/direction.

In mathematics, Fourier analysis of a periodic function is used to analyze the daily T2m at each grid $(i, j)$ as follows:

$$F_{i,j}(t) = a_{0,i,j} + \sum_{n=1}^{N}\left\{ a_{n,i,j} \cos\left(\frac{2\pi n t}{T}\right) b_{n,i,j} \sin\left(\frac{2\pi n t}{T}\right)\right\}, \tag{2}$$

where $T$ is the one-year period, $t$ is the time, and $a_n$ and $b_n$ are the Fourier coefficients. In the present case (seasonal cycle is dominant), only the terms up to $n = 1$ can be kept. Thus, Equation (3) can be written in the following form:

$$F_{i,j}(t) = a_{0,i,j} + a_{1,i,j}\cos\left(\frac{2\pi t}{T}\right) + b_{1,i,j}\cos\left(\frac{2\pi t}{T}\right) = a_{O,i,j} + A_{1,i,j}\cos\left(\frac{2\pi t}{T} + \varphi_{i,j}\right) \tag{3}$$

where $A$ is the amplitude and $\varphi$ is the phase angle.

(b)    Probability density

Finally, the histogram is used to display the probability density of hourly data values by dividing up the full range of the data into a small number of intervals (bins equal to 1 °C for daily T2m or equal to 1 m/s for daily wind speed). In the current study, the horizontal axis represents the possible range of the daily data values, and the vertical axis represents the frequency of occurrence [%].

## 3. Results

### 3.1. RegCM-SVN Verification

#### 3.1.1. Spatial Verification

Simulated RegCM-SVN results of T2m and surface wind field for the study area were examined to evaluate the accuracy of RegCM-SVN simulation compared with the ERA5 database. The spatial correlation between daily mean modeled simulation and the daily mean of ERA5 are shown in Figure 2. Furthermore, daily spatial bias (modeled simulation minus ERA5) is also shown in Figure 2.

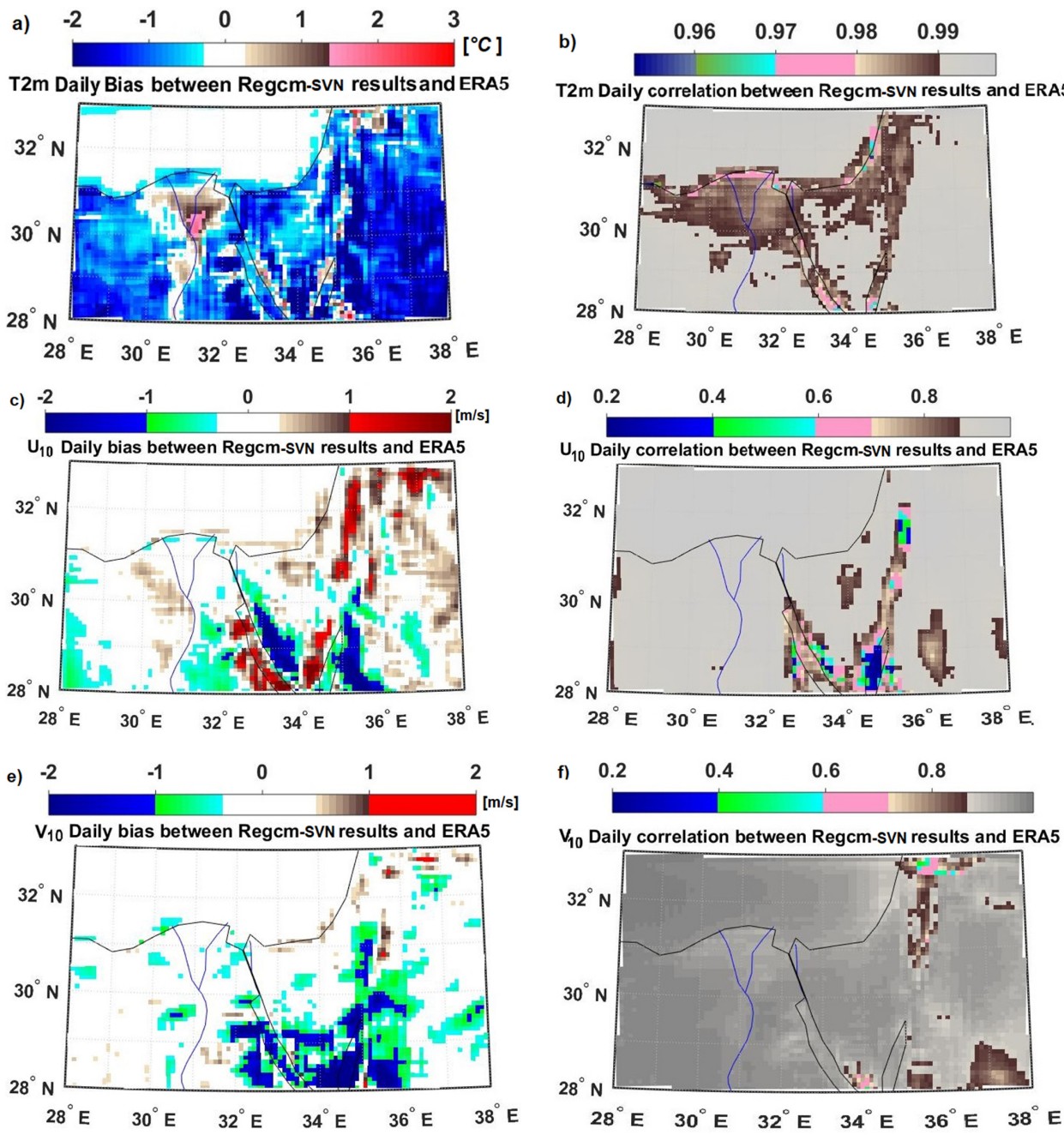

**Figure 2.** Verification analyses (bias and correlation) between RegCM-SVN and ERA5 database over the Southeastern Levantine. Positive bias means that RegCM-SVN shows an overestimation of the studied parameters compared with ERA5.

The simulated T2m results showed good agreement when compared with T2m of ERA5 over the SEL basin (Mediterranean part of the study area) and along the River Nile (vegetation area). The T2m bias between RegCM-SVN and ERA5 includes non-significant values over 35% of the study area, especially over SEL Basin and along the River Nile, as shown in Figure 2a. Over the southern Nile Delta, the T2m RegCM-SVN overestimated T2m ERA5 by about 1.1 °C on average. However, RegCM-SVN underestimated T2m over the Eastern Egyptian Desert, Northwestern Egypt, and over Sinai and the North-East part of Saudi Arabia Kingdom (area with mountains) by about 1.25 °C on average (Figure 2a).

On the other hand, the daily simulated time series of T2m (1979–2018) showed a strong correlation (over 0.96) with ERA5 T2m over the entire study area (Figure 2b). Thus, the results of RegCM-SVN proved to adequately simulate T2m over oceanic and vegetation

areas than over areas of desert or mountain features. This variation may be attributed to albedo computation in the RegCM-SVN simulation. RegCM-SVN calculations of surface albedo for each grid points depending on the land use characteristics are described in detail in Dickinson et al. [32]. For example, wetness and soil type were important for the bare soil. In general, the obtained T2m results indicated that the RegCM-SVN can be used to simulate T2m appropriately and efficiently over the study area.

Over 60% of the study area, especially over the SEL basin, Nile Delta, and along the River Nile, the $U_{10}$ simulated by RegCM-SVN showed a non-significant bias with the $U_{10}$ ERA5 data (Figure 2c). Moreover, RegCM-SVN overestimated $U_{10}$ by about 1.2 m/s along the eastern side of the Gulfs of Aqaba and Suez. Conversely, RegCM-SVN underestimated $U_{10}$ by about 1.3 m/s along the western side of the Gulfs of Aqaba and Suez. On the other hand, the daily time series of simulated $U_{10}$ had a higher correlation (>0.85) with daily ERA5 over 85% of the study area, most markedly over the SEL basin (Figure 2d). This correlation became weaker along the Gulfs of Aqaba and Suez (it has a mountainous nature). Thus, RegCM-SVN proved to simulate $U_{10}$ over the study area in a reasonable way. To increase $U_{10}$ simulation accuracy, RegCM-SVN should improve the calculation methods over mountainous areas.

There was also a non-significant $V_{10}$ bias between RegCM-SVN and ERA5 over 78% of the study area, especially over the SEL basin. In addition, RegCM-SVN underestimated $V_{10}$ along the Gulfs of Aqaba and Suez, partly owing to the mountainous nature along these gulfs. On the other hand, the daily $V_{10}$ simulations had a strong correlation (>0.85) with daily ERA5 over 96% of the study area, markedly over the SEL basin. This significant correlation decreases along the Gulfs of Aqaba and Suez.

According to the fact that topography controls the accuracy of the regional climate models [36], the mountainous nature (complexity of their topography) along the Gulfs of Aqaba and Suez is responsible for the low accuracy over the mountain regions in comparison with the rest of the study area. This may be explained by the resolution used, which is too crude to adequately represent the topographic features of the mountain regions [37]. Moreover, the RegCM-SVN simulations for $V_{10}$ showed a better agreement (lower bias and higher correlation) with the ERA5 than RegCM-SVN simulations than $U_{10}$ did over these mountains, as seen in Figure 2b–e. This may be explained by the shape of the nearby mountains [38], where these mountains stretch longitudinally, not latitudinally.

### 3.1.2. Verification Using Atmospheric Observations over the Five Studied Stations

To evaluate the performance of RegCM-SVN in simulating the atmospheric parameters on a station level, a comparison of RegCM-SVN simulations with observations/ERA5 over the five studied weather stations covering the overlapped period from 2007 to 2018 was conducted (Tables 2–7, Figures 3–5).

**Table 2.** The correlations of surface air temperature between observed, ERA5, and RegCM-SVN data sets from 2007 to 2018 after (before) removing the seasonal cycle at the five weather stations. The identification numbers (IN) 1 to 5 are also included in relation to Figure 1. Values between the brackets for the correlation after removing seasonal cycle. For more statistics, see Tables 3 and 4.

| Stations | | Ras El Tin {1} | Abu Qir {2} | Port Said {3} | El Arish {4} | Cairo airport {5} |
|---|---|---|---|---|---|---|
| | | Obser. | Obser. | Obser. | Obser. | Obser. |
| **Surface air temperature** | **ERA5** | 0.97 (0.85) | 0.96 (0.78) | 0.97 (0.87) | 0.96 (0.88) | 0.57 (0.45) |
| | **RegCM-SVN** | 0.96 (0.84) | 0.92 (0.76) | 0.95 (0.85) | 0.95 (0.84) | 0.85 (0.65) |

**Table 3.** Root mean square error (RMSE; calculated based on Equation (1) of the difference between the observation from one side and ERA5/RegCM-SVN data sets for the other side from 2007 to 2018 at the studied five weather stations for surface air temperature (°C). The identification numbers (IN) 1 to 5 are also included in relation to Figure 1.

| Stations | | Ras El Tin {1} | Abu Qir {2} | Port Said {3} | El Arish {4} | Cairo airport {5} |
|---|---|---|---|---|---|---|
| | | Obser. | Obser. | Obser. | Obser. | Obser. |
| Normal condition | ERA5 | 1.32 | 1.34 | 1.37 | 1.88 | 5.87 |
| | RegCM-SVN | 1.38 | 1.32 | 1.42 | 1.79 | 2.02 |
| Extreme condition | ERA5 | 3.33 | 3.10 | 3.32 | 2.82 | 1.95 |
| | RegCM-SVN | 2.83 | 1.75 | 2.64 | 1.41 | 1.29 |

**Table 4.** Surface air temperature (°C) annual statistics at Ras El Tin, Abu Qir, Port Said, El Arish, and Cairo airport (St. dv refers to standard deviation). All statistics were calculated based on daily data from 2007 to 2018. The identification numbers (IN) 1 to 5 are also included in relation to Figure 1; numbers of observations are showed in Table 1.

| Statistics | Annual Mean | | | Skewness | | | St. dv | | | Min | | | Max | | |
|---|---|---|---|---|---|---|---|---|---|---|---|---|---|---|---|
| Model Station | Observed | ERA5 | RegCM-SVN | Observed | ERA5 | RegCM-SVN | Observed | ERA5 | RegCM-SVN | Observed | ERA5 | RegCM-SVN | Observed | ERA5 | RegCM-SVN |
| Ras El Tin {1} | 21.9 | 21.1 | 20.8 | −0.09 | −0.07 | −0.08 | 4.81 | 4.45 | 4.29 | 5.20 | 8.19 | 8.09 | 41.00 | 33.67 | 36.29 |
| Abu Qir {2} | 21.5 | 21.1 | 21.1 | −0.11 | −0.08 | −0.05 | 4.76 | 4.69 | 5.74 | 7.40 | 7.00 | 4.05 | 38.00 | 36.73 | 43.83 |
| Port Said {3} | 22.1 | 21.6 | 20.5 | −0.16 | −0.14 | −0.19 | 5.35 | 4.81 | 4.71 | 1.00 | 7.56 | 6.37 | 40.70 | 37.62 | 42.79 |
| El Arish {4} | 21.6 | 21.2 | 19.9 | −0.19 | −0.15 | −0.10 | 5.99 | 4.95 | 6.22 | 2.00 | 6.94 | −0.54 | 45.00 | 35.71 | 43.39 |
| Cairo airport {5} | 23.1 | 22.1 | 23.7 | 0.06 | 0.04 | −0.04 | 6.85 | 7.36 | 7.79 | 5 | 2.99 | 2.55 | 45 | 45.08 | 45.96 |

**Table 5.** The correlations of surface wind speed between observed, ERA5, and RegCM-SVN data sets from 2007 to 2018 after (before) removing the seasonal cycle at the five weather stations (the identification numbers (INs) 1 to 5 are also included in relation to Figure 1). Values between the brackets for the correlation after removing seasonal cycle (for more statistics, see Tables 6 and 7).

| Stations | | Ras El Tin {1} | Abu Qir {2} | Port Said {3} | El Arish {4} | Cairo Airport {5} |
|---|---|---|---|---|---|---|
| | | Obser. | Obser. | Obser. | Obser. | Obser. |
| Surface wind speed | ERA5 | 0.74 (0.71) | 0.75 (0.74) | 0.70 (0.69) | 0.65 (0.65) | 0.06 (0.03) |
| | RegCM-SVN | 0.72 (0.70) | 0.75 (0.74) | 0.59 (0.58) | 0.49 (0.48) | 0.68 (0.67) |

**Table 6.** Root mean square error (RMSE; calculated based in Equation (1) of the difference between the observation from one side and ERA5/RegCM-SVN data sets for the other side from 2007 to 2018 at the five studied weather stations for surface wind speed (m/s) (the identification numbers (INs) 1 to 5 are also included in relation to Figure 1).

| Stations | | Ras El Tin {1} | Abu Qir {2} | Port Said {3} | El Arish {4} | Cairo airport {5} |
|---|---|---|---|---|---|---|
| | | Obser. | Obser. | Obser. | Obser. | Obser. |
| Normal condition | ERA5 | 1.92 | 2.21 | 1.66 | 1.90 | 2.62 |
| | RegCM-SVN | 1.99 | 2.01 | 1.99 | 1.75 | 1.47 |
| Extreme condition | ERA5 | 2.23 | 2.72 | 2.08 | 1.88 | 1.10 |
| | RegCM-SVN | 1.48 | 2.70 | 2.05 | 1.55 | 0.61 |

**Table 7.** Wind speed (m/s) annual statistics for observed, ERA5, and RegCM- SVN data sets at Ras El Tin, Abu Qir, Port Said, El Arish, and Cairo airport (St. dv refers to standard deviation). All statistics are calculated based on daily data from 2007 to 2018 (the identification numbers 1 to 5 are also included in relation to Figure 1; numbers of observations are shown in Table 1).

| Statistics | Annual mean | | | Skewness | | | St. Dev. | | | Min | | | Max | | |
|---|---|---|---|---|---|---|---|---|---|---|---|---|---|---|---|
| Model Station | Observed | ERA5 | RegCM-SVN | Observed | ERA5 | RegCM-SVN | Observed | ERA5 | RegCM-SVN | Observed | ERA5 | RegCM-SVN | Observed | ERA5 | RegCM-SVN |
| Ras El Tin {1} | 5.44 | 5.06 | 5.35 | 0.79 | 0.80 | 0.77 | 2.83 | 2.13 | 2.32 | 0.00 | 0.05 | 0.00 | 24.84 | 18.55 | 21.08 |
| Abu Qir {2} | 5.88 | 4.52 | 4.88 | 0.74 | 0.82 | 0.82 | 2.66 | 1.88 | 2.03 | 0.00 | 0.01 | 0.01 | 24.30 | 16.70 | 18.04 |
| Port Said {3} | 4.92 | 4.73 | 5.11 | 0.21 | 0.56 | 0.46 | 2.21 | 2.00 | 2.18 | 0.00 | 0.04 | 0.01 | 21.60 | 15.73 | 18.10 |
| El Arish {4} | 4.28 | 4.20 | 3.82 | 0.59 | 0.96 | 0.96 | 2.41 | 2.14 | 2.15 | 0.00 | 0.04 | 0.53 | 27.00 | 17.57 | 18.09 |
| Cairo airport {5} | 4.13 | 3.76 | 2.80 | 0.39 | 0.65 | 0.48 | 2.18 | 1.66 | 1.86 | 00 | 0.02 | 00 | 16.74 | 14.4 | 14.16 |

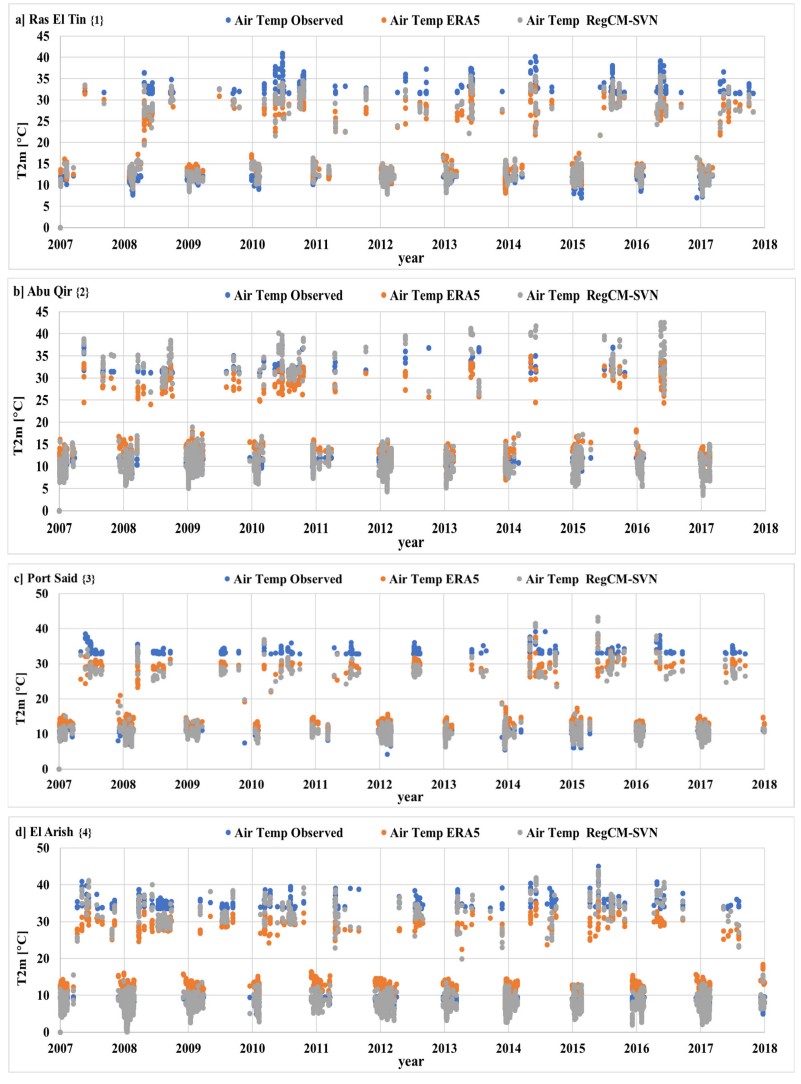

**Figure 3.** *Cont.*

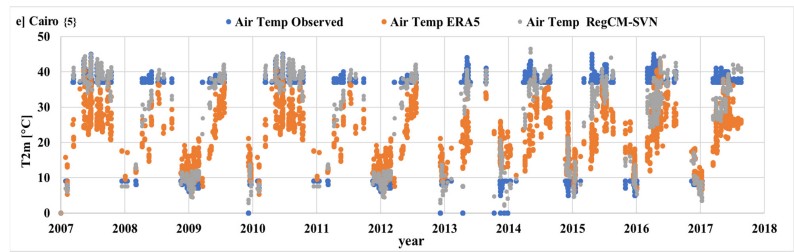

**Figure 3.** Hourly extreme time series of surface air temperature for observed, ERA5, and RegCM-SVN at Ras El Tin (**a**), Abu Qir (**b**), Port Said (**c**), El Arish (**d**), and Cairo airport (**e**). Only extreme values of observed data and its corresponding ERA5 and RegCM-SVN are presented. The identification numbers 1 to 5 are also included in relation to Figure 1.

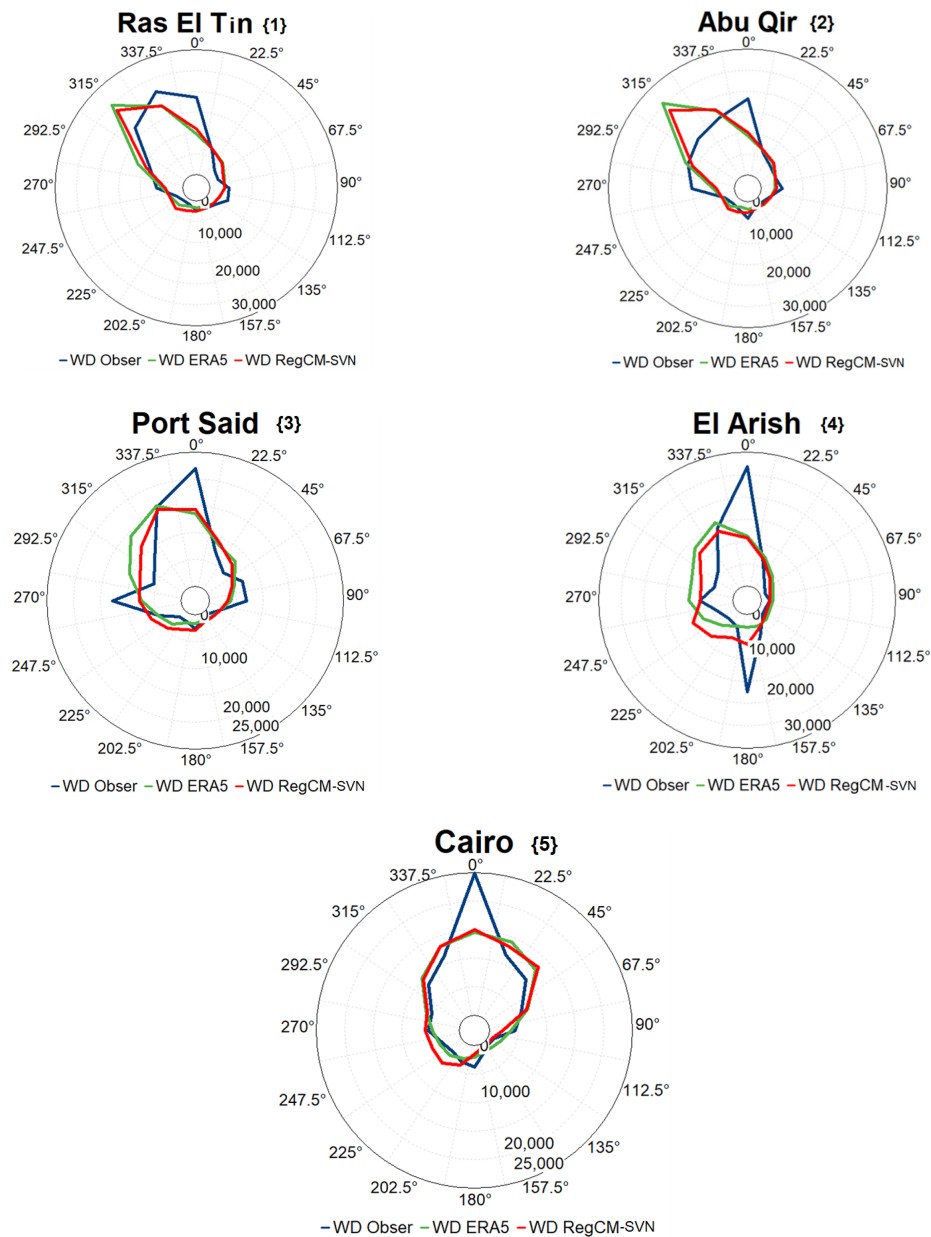

**Figure 4.** Wind rose for the observed ERA5 and RegCM-SVN result at Ras El Tin, Abu Qir, Port Said, El Arish, and Cairo airport. The occurrence is also shown in the figure (the identification numbers 1 to 5 are also included in relation to Figure 1; WD refers to wind direction; Obser refers to observation).

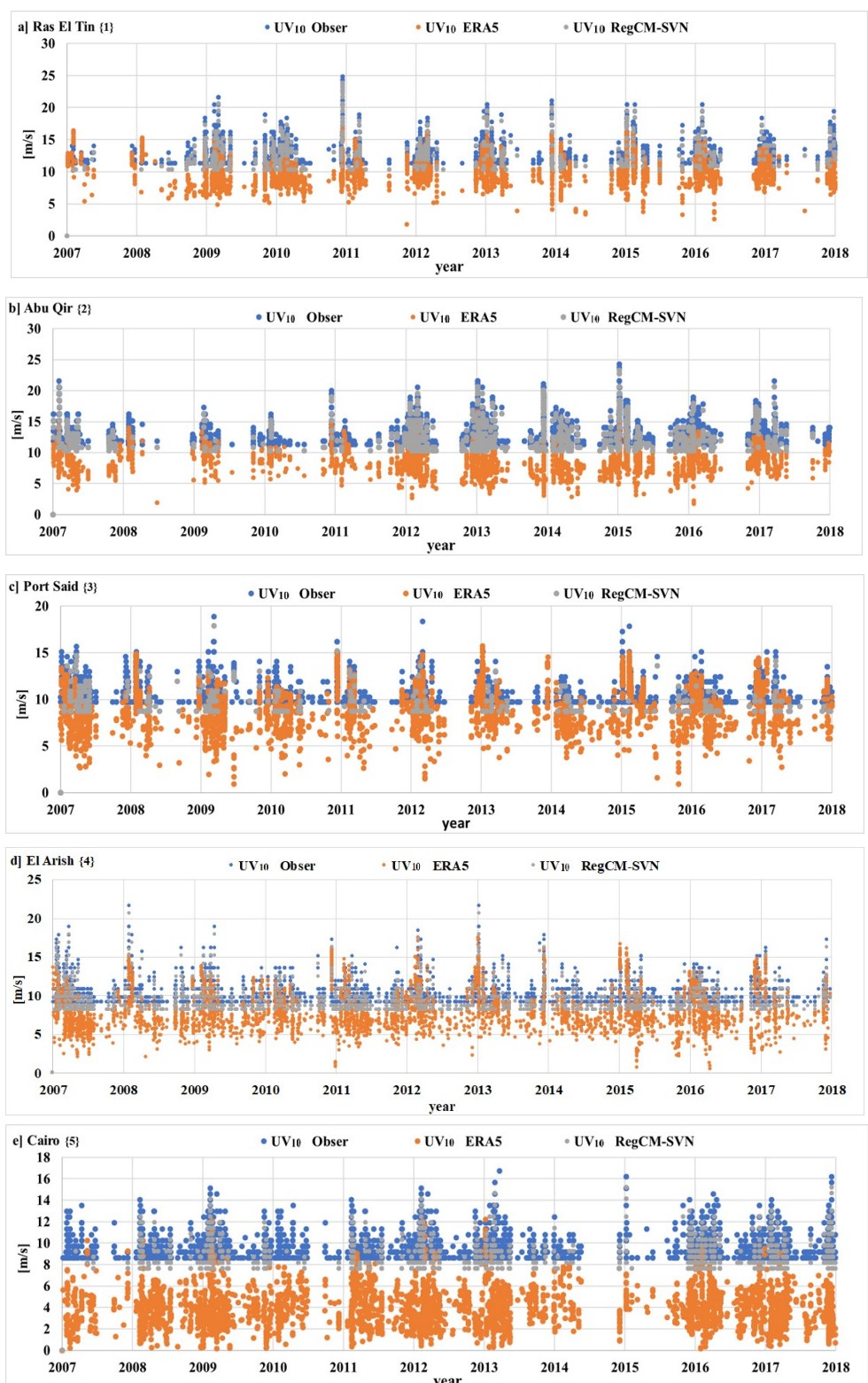

**Figure 5.** Hourly extreme time series of surface wind speed for observed, ERA5, and RegCM-SVN at Ras El Tin (**a**), Abu Qir (**b**), Port Said (**c**), El Arish (**d**), and Cairo airport (**e**). The only extreme values of observed data and their corresponding ERA5 and RegCM-SVN results are presented (the identification numbers 1 to 5 are also included in relation to Figure 1).

(a)    Surface air temperature (T2m) verification

Hourly simulated surface air temperature by RegCM-SVN showed a significantly strong correlation with the corresponding observed T2m over the studied five stations before removing the seasonal cycle; ranging from 0.85 over Cairo airport to 0.96 over Ras El Tin, as shown in Table 2. After removing the seasonal cycle, the correlation coefficient between hourly RegCM-SVN T2m simulation and the corresponding observations still describes a significantly strong correlation over the studied stations, ranging from 0.65 over Cairo airport to 0.85 over Port Said. Furthermore, the RegCM-SVN shows a similar accuracy to ERA5 in describing the observed T2m over four out of five studied stations based on correlation coefficient (Table 2) and RMSE values (Table 3). However, over Cairo airport, RegCM-SVN T2m simulation showed a significant improving in describing the observed T2m with respect to ERA5. Moreover, the simulated hourly T2m by RegCM-SVN showed a better capability than ERA5 in simulating observed extreme events (Figure 3 and Table 3) over the five studied stations. The RMSE of the difference between observation and RegCM-SVN is significantly less than 0.5 times the difference between observation and ERA5, as seen in Table 3. During the extreme observed events, the RegCM-SVN T2m values are so close to observed values compared with the difference between ERA5 and observed data (Figure 3).

In Table 4, it is shown that RegCM-SVN underestimated [overestimated] annual mean T2m (1979–2018) compared with the observed annual mean T2m over the only studied coastal [inland] stations by about 1 °C, 0.4 °C, 1.6 °C, and 1.7 °C [0.7 °C] over Ras El Tin, Abu Qir, Port Said, and El Arish respectively [Cairo airport]. Analyses of skewness coefficient showed that both simulated and observed T2m have almost symmetrical probability distribution with a small tendency towards negative skewness over the studied five stations. Similarly, the values of standard deviation (bias between RegCM-SVN and observation averaged 0.2 of the studied area) supported that RegCM-SVN closely simulated the observed air temperature over the five studied stations. It is clear that the standard deviation of RegCM-SVN T2m simulation has deviated in quite different directions from the observed data (Table 4). Over Ras El Tin and Port Said (Abu Qir, El Arish, and Cairo airport), the standard deviation calculated from RegCM-SVN simulation showed lower (higher) values than that calculated from the observed data. This may be related to the data range (maximum minus minimum); RegCM-SVN T2m simulation has a narrow (wide) range compared with that observed over Ras El Tin and Port Said (Abu Qir, El Arish, and Cairo airport). RegCM-SVN overestimated (underestimated) the observed minimum value of the annual T2m mean over Ras El Tin and Port Said (Abu Qir, El Arish, and Cairo airport). Finally, RegCM-SVN overestimated the observed maximum value of the annual T2m mean compared with observed data over Abu Qir, Port Said, and Cairo airport by 5.8 °C, 2.1 °C, and 1 °C, respectively, and gave a lower estimate for maximum value of the annual T2m mean compared with observations over Ras El Tin and El Arish by 4.7 °C and 1.6 °C, respectively.

(b)    Wind direction verification

At Ras El Tin, RegCM-SVN and ERA5 data indicated a predominant wind direction from NW. However, the observed data indicated a predominant NNW direction with a clockwise deviation of 22.5° from the RegCM-SVN and ERA5 results. Over Abu Qir, RegCM-SVN and ERA5 pointed towards the same WD pattern with a predominant NW wind direction; however, the observations showed a predominant N wind direction with 45° clockwise deviations from the RegCM-SVN and ERA5 results. At Port Said and El Arish stations, RegCM-SVN and ERA5 showed the same predominant NNW wind direction; however, the observed wind direction was predominantly blowing from N with 22.5° clockwise deviation from the RegCM-SVN and ERA5 results. At Cairo airport, the data showed a similar pattern, where the wind blows dominantly from the N (14.9% of the time) and NNE (14.9% of the time); however, the observation showed that N is the predominant direction (24.9% of the time as seen in (Figure 4). Only over the

El Arish station, the observed wind blew from the S direction, describing the second permanent wind direction; however, over the other four studied stations, the first and second permanent wind direction blew from N origin directions. However, this significant southerly wind over El Arish station was not observed by the RegCM-SVN or ERA5 results, partly showing the importance of using much a higher resolution (*maybe* $\frac{1}{3}$ of the used grid size (=3.3 km) or a higher resolution than 3.3 km) of RegCM-SVN simulations. The use of a much higher resolution in downscaling simulations is expected to improve the simulations results [39,40]. In general, there was a good similarity between the performance of RegCM-SVN and ERA5 over the five stations. Moreover, the RegCM-SVN and ERA5 results of wind direction were closely related to the observations over Ras El Tin, Port Said, El Arish, and Cairo airport rather than Abu Qir.

(c) Wind speed ($UV_{10}$) verification

RegCM-SVN hourly $UV_{10}$ simulation showed a strong significant correlation with the corresponding observed values over the five studied stations before removing the seasonal cycle; ranging from 0.49 over El Arish to 0.75 over Abu Qir, as shown in Table 5. After removing the seasonal cycle, the correlation coefficient between hourly RegCM-SVN $UV_{10}$ simulation and the corresponding observations showed non-significant difference before and after the removal of the seasonal cycle, which indicates that the seasonal cycle has no significant effect on the strong relation between hourly RegCM-SVN $UV_{10}$ simulation and the corresponding observations. There is a low correlation value between ERA5 and observed data over Cairo airport after (before) removing the seasonal cycle equal to 0.06 (0.03). Although the annual mean and standard deviation bias between the observed and ERA5 are 0.96 m/s and 0.32 m/s respectively. This may indicate that ERA5 can describe the observed wind speed over Cairo airport on a monthly or even longer scale, but on an hourly scale, ERA5 poorly describes the Cairo airport observed wind speed. This low correlation between ERA5 and the observed data over Cairo airport suggested a significant correlation between the RegCM-SVN UV10 simulation and observed UV10 data over Cairo airport. This shows how the RegCM-SVN UV10 simulation describes the observed value more accurately over Cairo airport.

In general, RegCM-SVN underestimates the annual mean wind speed (1979–2018) against the observed values at Ras El Tin, Abu Qir, El Arish, and Cairo airport by 0.1 m/s, 1 m/s, 0.5 m/s, and 1.3 m/s, respectively. At the same time, RegCM-SVN showed a higher estimate for annual mean observed wind speed at Port Said by 0.2 m/s. Generally, the maximum values of the annual average wind speed at the studied stations calculated by the RegCM-SVN results are lower than those calculated using the observed values, as given in Table 7.

According to the skewness coefficient, standard deviation, and minimum value, the RegCM-SVN simulation results showed close statistical measures to their corresponding observed wind speed results (Table 7).

Finally, it is clear from Table 7, where annual mean, skewness, standard deviation, and minimum and maximum values of RegCM-SVN, ERA5, and observations are so close to each other, such that RegCM-SVN shows a similar accuracy to ERA5 when compared with observed wind speeds in general. Moreover, at extreme hourly events, the simulated $UV_{10}$ by RegCM-SVN shows better accuracy in reflecting the observed situations than ERA5, where the RegCM-SVN $UV_{10}$ values are very close to the observed values compared with the difference between ERA5 and the observed data (Figure 5). In detail and according to RMSE analyses (Equation (1)), the RegCM-SVN simulations displayed better accuracy than ERA5 in describing hourly observed $UV_{10}$ extreme events (Table 6), where the RMSE of the difference between observation and RegCM-SVN is significantly less than 0.75 times the difference between observation and ERA5. As expected, the verification utilizing the five weather stations reveals certain biases of T2m and wind regime (speed and direction). This may be because of some possible reasons. The first reason may be related to the RegCM-SVN physics (e.g., Albedo or advection scheme calculation). The second reason may be related to the boundary condition (ERA5). The third reason is related to the global

land cover characteristics used (grid size ≃ 1 km). The authors believe that the boundary condition can be improved using some observed station alongside ERA5.

### 3.2. Spatial and Temporal Distribution of T2m, Wind Speed, and Direction over SEL

The annual mean T2m (from 1979 to 2018) simulated by RegCM-SVN ranged from 19 °C to 20 °C for the western sector of the SEL Basin, while it ranged from 20 °C to 21 °C over the eastern sector of the SEL basin (Figure 6a). This finding showed a range of 2 °C in the annual mean T2m (from 1979 to 2018), while a range of 1 °C was reported by Shaltout et al. [16] over the SEL basin (1979–2010). This difference may be related to the use of the regional climate model (current finding) instead of the global climate model that was used previously by Shaltout et al. [16], as well as to the different time span. However, the range of the annual mean T2m (from 1979 to 2018) over the land area (14–26 °C) of SEL is close to that calculated by Almazroui [41].

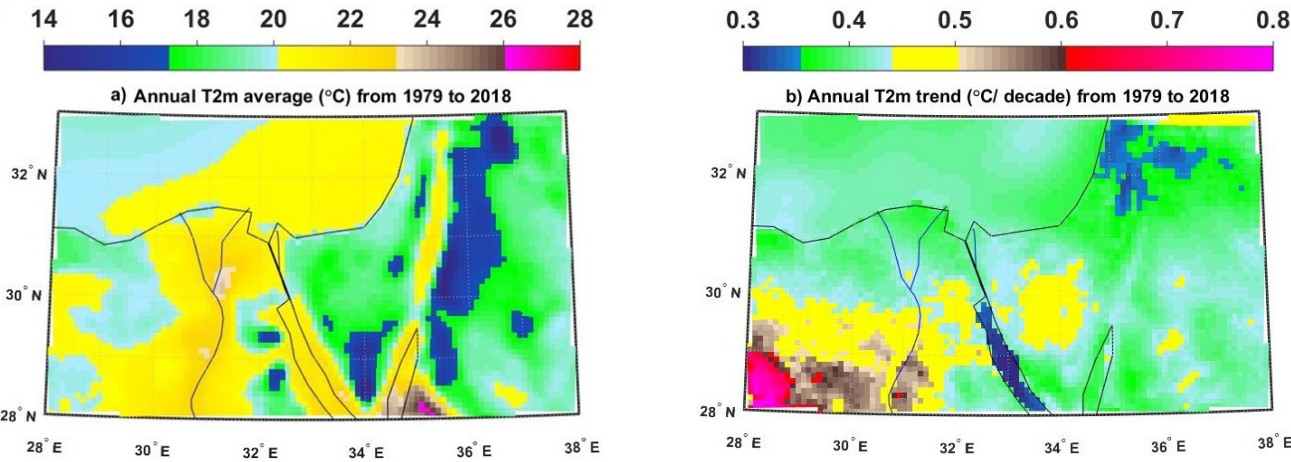

**Figure 6.** Spatial and temporal distribution of the mean annual SEL for the simulated T2m by RegCM-SVN over the period from 1979 to 2018.

Over the Delta region and Gulfs of Aqaba and Suez, the annual mean T2m ranged from 20 °C to 21 °C. On the other hand, the annual mean T2m for the same period of study reached its maximum values (23–26 °C) at the southeastern side of the Gulf of Aqaba. In addition to that, in between the two Gulfs, the annual mean T2m reached its minimum values from 14 °C to 18 °C, which were related to the mountain nature (Saint Katherine mountain) and, similarly, over El-Jalala mountain, as shown in Figure 6a. The mountain nature affects the RegCM-SVN simulation in two ways, the mountain elevation that is not fully accurately calculated by the used Global Land Cover Characteristics (30 arc seconds) was the first, while the complexly of the induced local circulation around the mountain was the second. In general, the annual mean of T2m (from 1979 to 2018) over Sinai Peninsula increased from south to north, partly owing to the mountain nature over its southern part.

More than 70% of the study area had an annual T2m trend rate (from 1979 to 2018) that ranged from 0.35 °C/decade to 0.45 °C/decade, especially over the SEL basin, Gulf of Aqaba, Nile Delta, and the northern Egyptian coast. Moreover, the annual T2m trend (from 1979 to 2018) reached its minimum trend rate value (0.30–0.32 °C/decade) over the Gulf of Suez. However, the annual T2m trend (from 1979 to 2018) reached its maximum values (>0.6 °C/decade) over the southwestern corner of the study area (Western Desert), as shown in Figure 6b. Moreover, on the Sinai Peninsula, the annual T2m (from 1979 to 2018) showed its highest trend rate over central Sinai (0.45–0.55 °C/decade), indicating that the T2m spatial distribution over Sinai may change in the future, where the highest temperature in Sinai may have occurred over its central part rather than northern part (current situation).

Over most of the SEL basin, the annual mean wind direction (from 1979 to 2018) simulated by RegCM-SVN showed 300°–330° (NW), except the east side of the SEL basin,

which displayed 270–300° (WNW), as shown in Figure 7a. Along the Gulf of Suez, the annual mean wind direction (from 1979 to 2018) showed the N direction in its northern part and the NW direction in its southern part, almost following the orientation of the Gulf of Suez. This may be explained by the mountain nature of this area, which allowed the wind to blow parallel to the mountain. The same concept can explain why the annual mean wind direction (from 1979 to 2018) along the Gulf of Aqaba indicated 10–30° following the coastal orientation.The annual mean wind speed (from 1979 to 2018) simulated by RegCM-SVN reached its maximum values (>8 m/s) over the western part of Gulfs of Suez and Aqaba, partly owing to the mountainous nature along these gulfs (land breeze, sea breeze, katabatic, and anabatic phenomena). This led the Egyptian Government to build wind power farms along the southwestern part of the Gulf of Suez. On the other hand, the annual mean wind speed (from 1979 to 2018) over the SEL basin ranged from 2 to 4 m/s, demonstrating an increasing pattern from east to west, as shown in Figure 7b. In over 60% of the study area, the annual mean wind speed (from 1979 to 2018) was less than 3 m/s; however, about 15% of the study area had greater than 4 m/s.

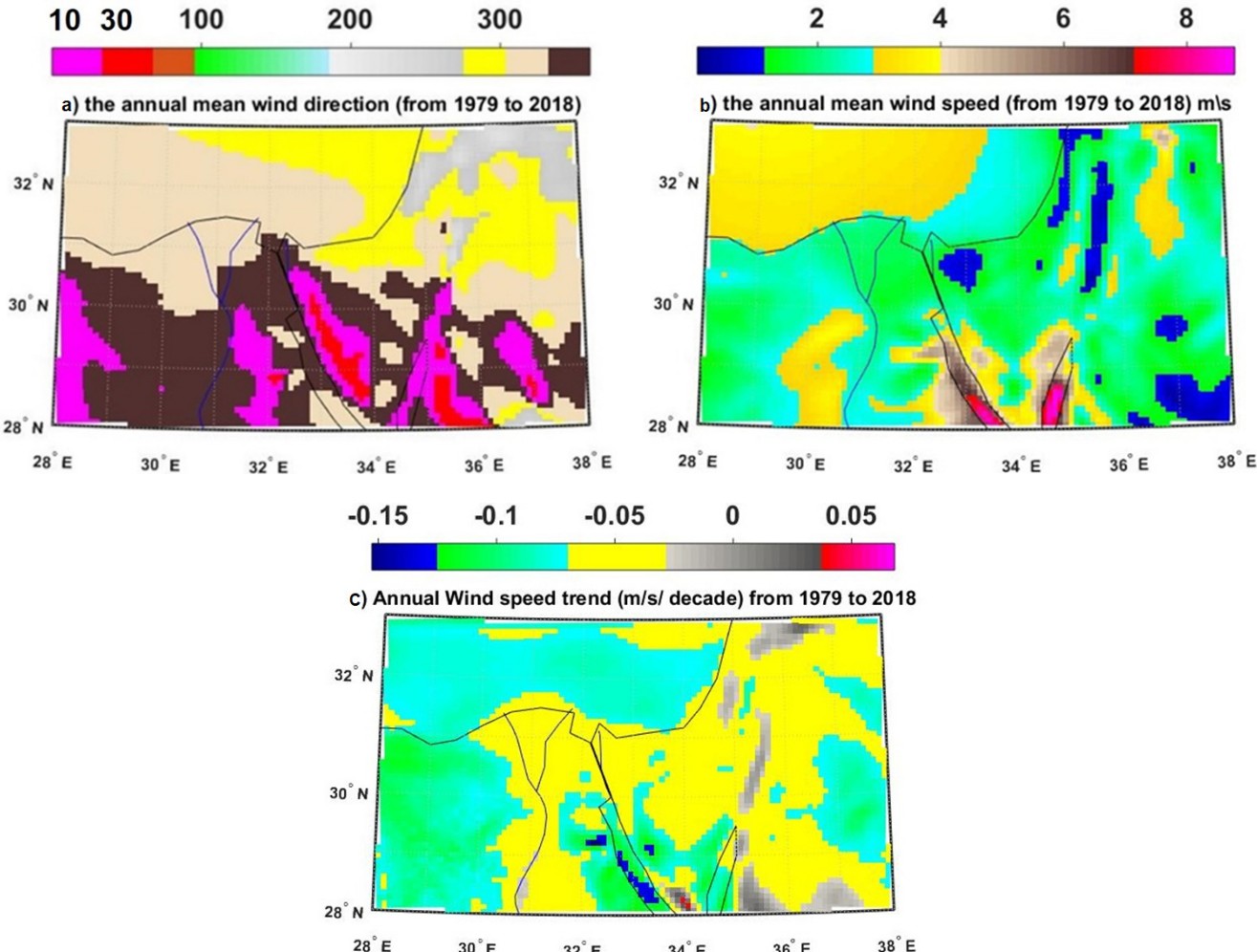

**Figure 7.** Spatial and temporal distribution of the mean annual SEL for the simulated wind speed and direction by RegCM-SVN over the period from 1979 to 2018.

In around 7% of the study area, especially over the Saudi Kingdom part, the annual trend rate of wind speed (from 1979 to 2018) simulated by RegCM-SVN gave insignificant (within the 95% confidence interval) values. Only about 1% of the study area (to the east of the southern part of the Gulf of Suez), the annual trend of wind speed (from 1979 to 2018) demonstrated positive trend values. Over 92% of the study area, the annual trend

of wind speed (from 1979 to 2018) gave negative trend values, most markedly along the Gulf of Suez (Figure 7c). Moreover, the change in aerodynamic roughness length should be considered to investigate this negative trend. Overall, this finding can give an early awareness about the sustainable development of wind power farm projects in that area, as the wind speed tends to show a decreasing trend.

### 3.3. Seasonal Characteristics of Surface Air Temperature and Wind Speed over SEL

Seasonal characteristics of both surface air temperature and wind speed were investigated along the study area during the period (from 1979 to 2018).

(i)     *Seasonal surface air temperature*

The warmest winters over the SEL (>15.96 °C; winter mean + double winter standard deviation) were observed during 2010 and 2018, while much cooler winters occurred in 1983 and 1992 (<11.91 °C; winter climatic mean–double climatic winter standard deviation) with a 95% confidence level, as shown in Figure 8.

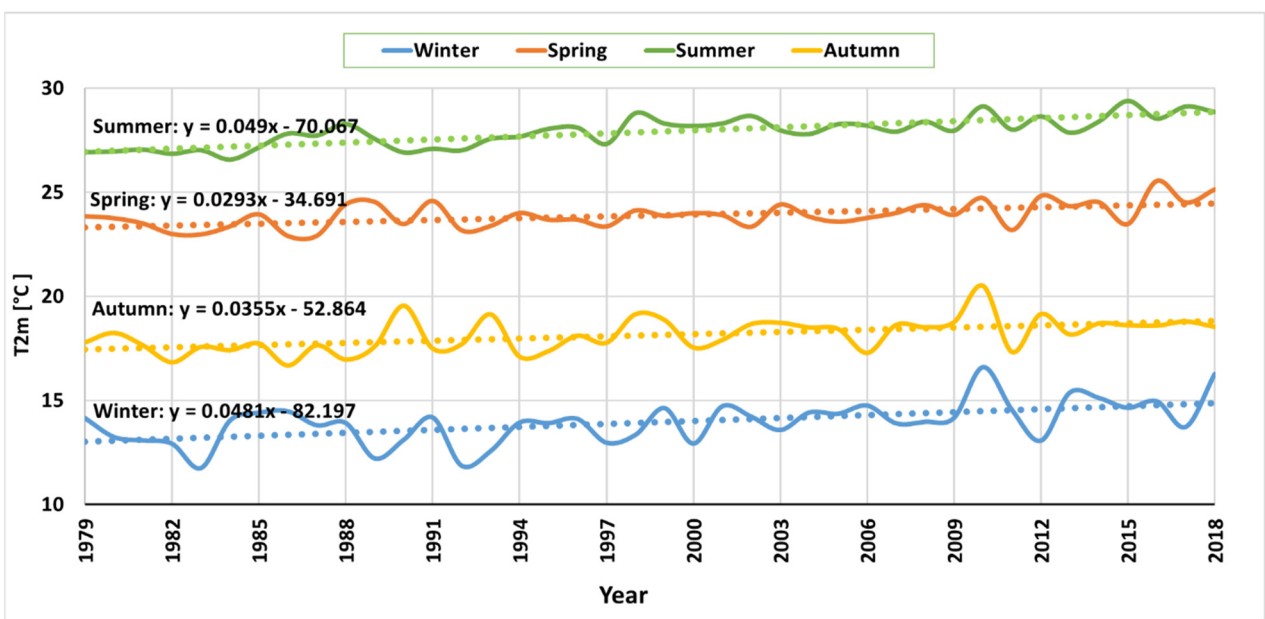

**Figure 8.** Seasonal 2 m air temperature (T2m) time series and linear trends (dotted line) calculate by RegCM-SVN over the study area. Linear regression equations for each season are presented on the left-hand side.

Concerning variability in summer seasons, years with much warmer summers over the SEL were observed during 2015 and 2010 (>29.3 °C; summer climatic mean + double climatic summer standard deviation) with a 95% confidence level, while much cooler summers occurred in 1984 (<26.5 °C; summer mean–double summer standard deviation).

The year 2010 showed the warmest year during winter and autumn seasons. During winter of 2010, the North Atlantic Oscillation Index (NAOI) showed a dramatic decrease in its value, as stated by Tsimplis et al. [42] and Mohamed et al. [43]. This low NAOI can explain warming events during 2010, as the negative NAOI phase is associated with warmer conditions over the Mediterranean [44]. During the wintertime, the negative NAOI phase is associated with a low-pressure cell that occurs over Iceland. This low-pressure cell arcs the jet stream south over the North Atlantic, funneling warm air to the Mediterranean Sea.

Surface air temperature displayed a significant seasonal variability, with average values varying from 13.9 °C during winter to 27.9 °C during summer, with linear increasing trends ranging from 0.29 °C/decade during spring to 0.49 °C/decade during summer, as shown in Figure 8. The highest warming trend rate in summer compared with the lower warming trend rate in spring partly indicated that the air temperature difference between

summer and spring seasons may increase in the future. On the contrary, the stronger warming trend in winter compared with the lower warming trend rate in autumn partly indicated that the air temperature difference between the winter and autumn seasons may decrease in the future. These probabilities of increasing the difference between summer and spring T2m together with decreasing the difference between the winter and autumn T2m are based on only 40 years' simulation, partially indicating that these two probabilities are not stable owing to the short climatic period used, which misses some variability on multi-decadal time scales, and the effect of sustained anthropogenic activity is not considered.

The annual variability of T2m reached its maximum value during the winter seasons and reached its minimum value during spring. These can be explained by the frequent occurrence of atmospheric depressions with associated fronts during winter. In general, all seasonal trend rates over the period (from 1979 to 2018) indicated an increasing trend, underlining the need for more downscaling simulation up to 2100 to confirm the regional warming trends.

As the T2m trends by season are very different (Figure 8), the T2m spatial distributions' trends by season are shown in Figure 9. There is a significant seasonal average warming trend (from 1979 to 2018) over SEL, ranging from 0.49 °C/decade during summer and spring to 0.33 °C/decade during autumn. During all seasons, the southwest corner of the study area describes the higher value of the seasonal trend. Moreover, the SEL basin has a lower value of T2m seasonal trend during autumn (0.25 °C/decade) and winter (0.27 °C/decade) than during summer (0.41 °C/decade).

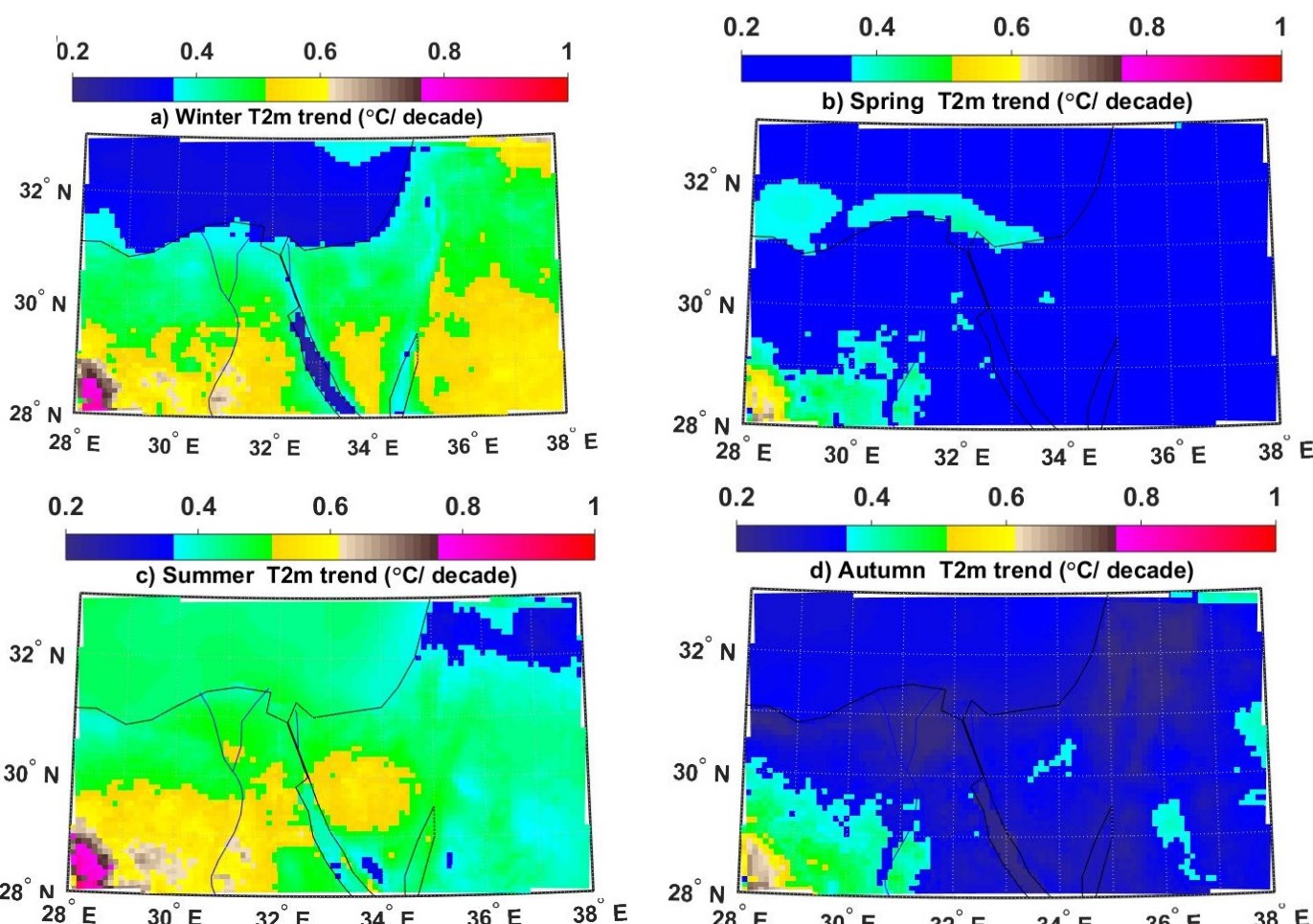

**Figure 9.** Spatial distribution of the mean seasonal T2m over the period from 1979 to 2018 over SEL.

(ii)  *Seasonal surface wind speed*

The windiest winters over SEL occurred during 1992 and 2012 with the conditions (>4.52 m/s; winter mean + double winter standard deviation), while the calmest winters occurred in 2014 (<3.65 m/s; winter mean–double winter standard deviation), as shown in Figure 10.

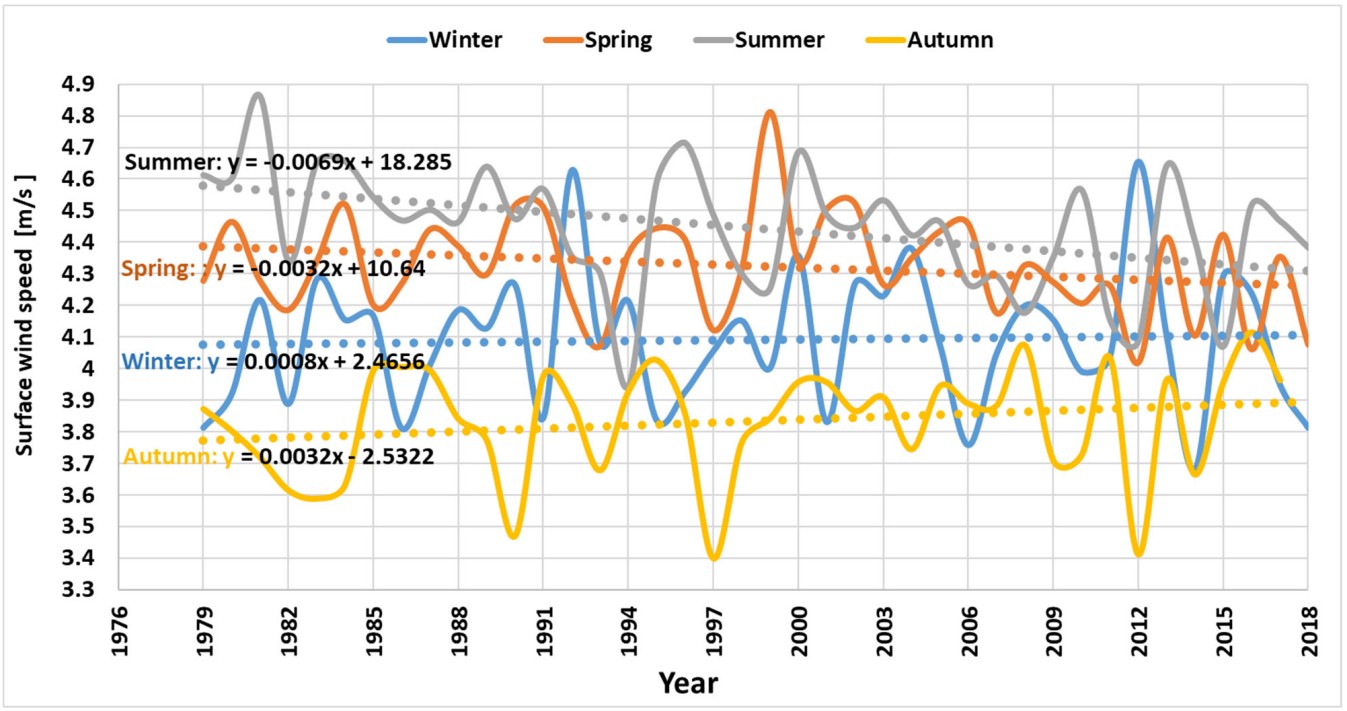

**Figure 10.** Seasonal surface wind speed time series and linear trends (dotted line) calculate by RegCM-SVN over the study area. Linear regression equations for each season are presented on the left-hand side.

For the summer season, the windiest summers over SEL occurred during the year 1981 (>4.82 m/s; summer mean + double summer standard deviation), while the calmest summer occurred in 1994 and 2015 (<4.06 m/s; summer mean–double summer standard deviation), with a 95% confidence level.

Surface wind speed markedly displayed seasonal variabilities, with mean values varying from 3.83 m/s during autumn to 4.44 m/s during summer, with increasing linear trends during autumn and winter together, while decreasing trends occurred during summer and spring, as shown in Figure 10. The trend rates of seasonal surface wind speeds suggested a possible decrease in wind speeds in the future in summer and spring.

The annual variability of wind speed reached its maximum value during the winter season, owing to the frequent occurrence of atmospheric winter depressions over the Mediterranean Sea [45], while reaching its minimum values during spring.

### 3.4. Variability of SEL Surface Air Temperature and Surface Wind

(a)  Seasonality analysis (Fourier analysis)

The seasonality analysis performed using Fourier analysis of 40 years of daily RegCM-SVN simulations for T2m showed a clear seasonality variation (seasonal cycle amplitude) ranging from 5 °C over the SEL basin to 11 °C over the eastern land part of the study domain, as shown in Figure 11a. The T2m seasonality is much larger over land than the Mediterranean Sea, owing to the higher specific heat capacity over the sea water. In addition, T2m seasonality showed a zonal increase from north to south over the SEL domain, partly indicating that the southern part has lower specific heat capacity (as the southern part is characterized by mountain or desert features).

The T2m relative phase over the SEL displayed a zonal variation ranging from nearly −105 days over the southern part of the study area to nearly −140 days over the SEL basin, indicating that the seasonal offset over the SEL domain is 35 days, as shown in Figure 11b. It is clear that the seasons begin later in the SEL basin, where the maximum in T2m occurred around 19 August, while the seasons come earlier over southern parts, where the maximum in T2m occurred around 15 July.

In general, the seasons come later in the southern part of the Gulf of Suez when compared with the Gulf of Aqaba and the northern part of the Gulf of Suez by about 5 days. This difference of 5 days in the phasing of the seasonal cycle be considered statistically significant as the actual shape of the annual cycle has a better approximation based on the use of a daily long time period. This finding supports the finding of Shaltout [34], who stated that the Gulfs of Aqaba and the northern part of the Gulf of Suez showed a similar relative phase, and their maximum sea surface temperature comes earlier by about 5 days than over the southern part of the Gulf of Suez.

(b)    Probability density

Over SEL, the most frequent T2m occurrences were 25–26 °C (6.7%), 24–25 °C (6.2%), 26–27 °C (5.7%), and 15–16 °C (5.3%), as shown in Figure 12. Over 75% of the time, T2m varied between 13 °C and 28 °C. However, over 98.3% of the time, T2m varied between 4 °C and 31 °C, indicating that T2m is rarely increased over 31 °C or decreased below 4 °C. Two existing peaks around 15 °C and 25 °C indicated that SEL can be divided into several clusters, as will be discussed in the future.

Over SEL, the most frequent wind speed occurrences were 3–4 m/s (20.5%), 2–3 m/s (17.7%), 4–5 m/s (17.6%), and 5–6 m/s (12.2%), as shown in Figure 13. Over 79.6% of the time, wind speed varied between 1 and 5 m/s. However, around 99% of the time, the wind speed varied between calm and 10 m/s, indicating that wind speed rarely exceeded 10 m/s.

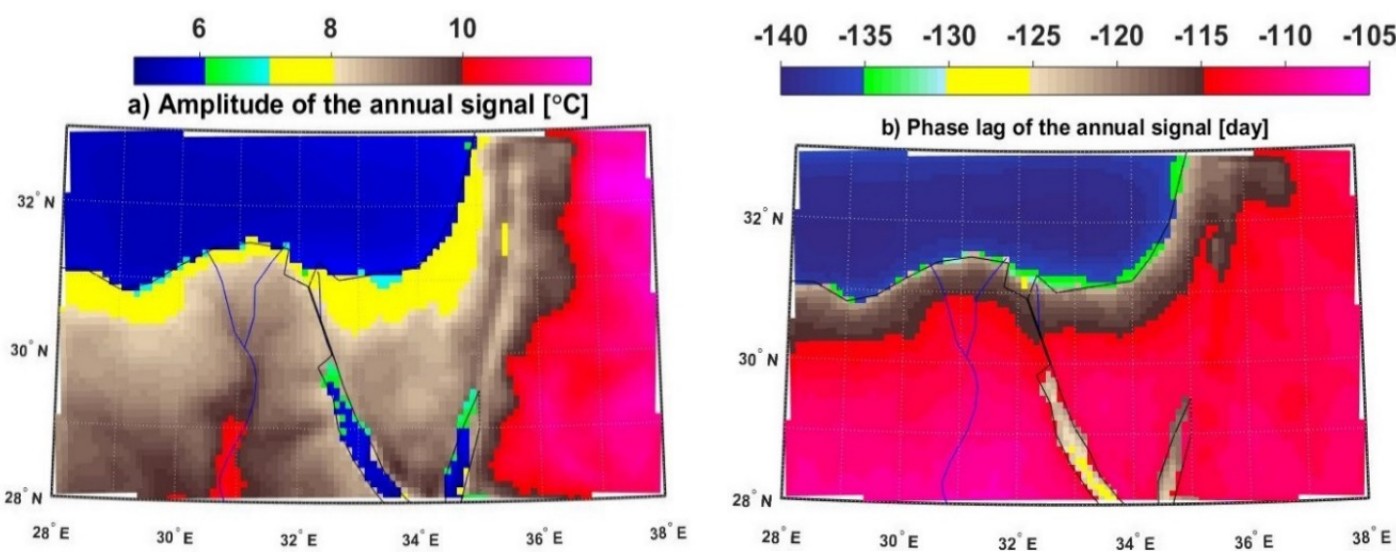

**Figure 11.** Spatial distribution of annual simulated surface air temperature signal: (**a**) amplitude and (**b**) relative phase, over the SEL area from 1979 to 2018.

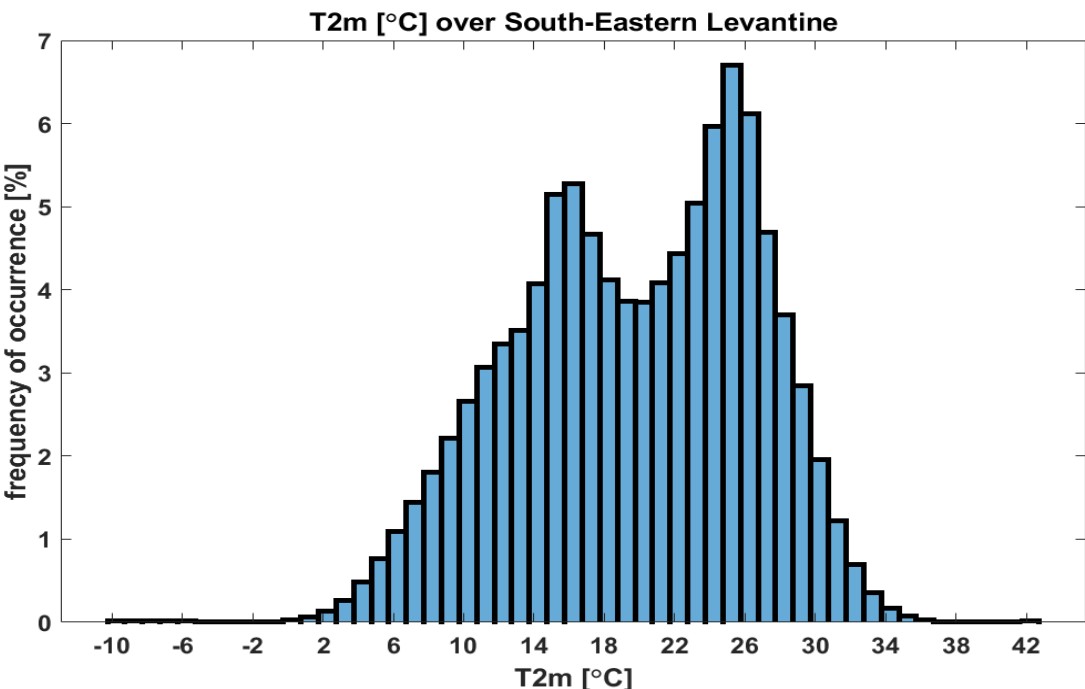

**Figure 12.** Histogram (probability density) of the daily surface air temperature (1979–2018) with a bin width of 1 °C over SEL (calculate by RegCM-SVN).

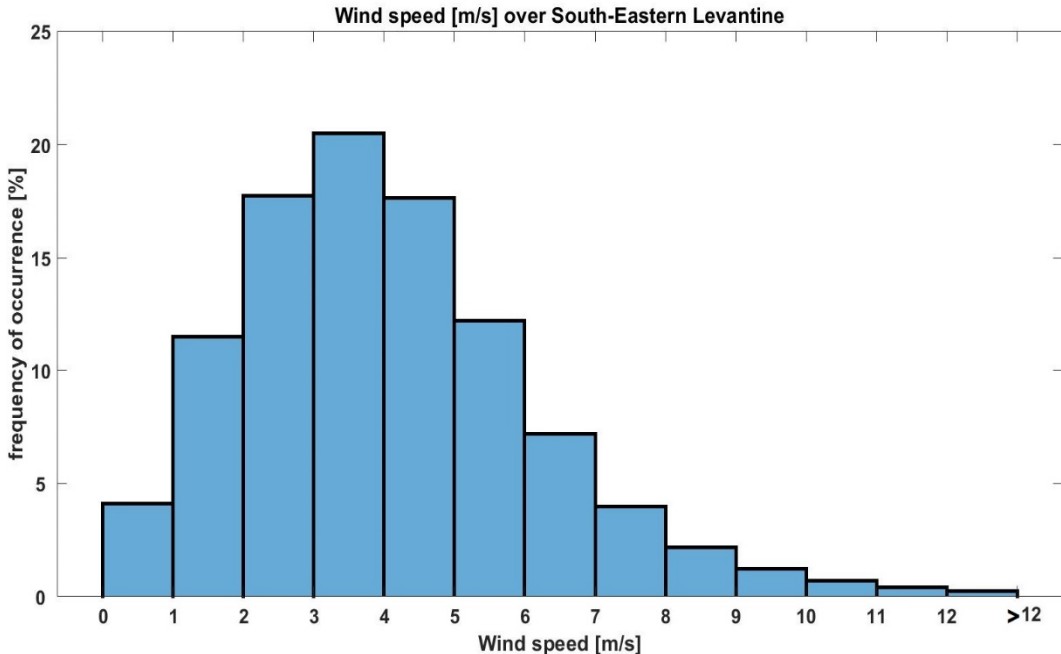

**Figure 13.** Histogram (probability density) of the daily surface wind speed (1979–2018) with a bin width of 1 m/s over SEL (calculate by RegCM-SVN).

## 4. Summary and Conclusions

The present work sheds light on the variations in surface air temperature and wind regime over the southeastern Levantine Basin and its surrounding areas. The current analysis was based on RegCM-SVN simulation results during the period 1979–2018.

The validation of RegCM-SVN results was first done against the ERA5 reanalysis (cell-to-cell validation) during the period 1979–2018 and then with observed data (using

the nearest neighbor algorithm to select the nearest grid to the observed station) during the period 2007–2018. Validation processes indicated that RegCM-SVN successfully simulated the surface air temperature and wind field over the study area. Generally, RegCM-SVN showed a similar accuracy to ERA5 in describing the surface wind field and T2m over the study area from 2007 to 2018. Moreover, and by comparing RegCM-SVN/ERA5 with observations during hourly extreme conditions using RMSE analyses, it can be concluded that RegCM-SVN showed better accuracy than ERA5 in simulating $UV_{10}$ and T2m during extreme events. Furthermore, wind rose analyses showed that downscaling of ERA5 data (using RegCM-SVN) significantly improves the calculations of wind direction. Thus, the use of RegCM-SVN in its current configuration is a better tool than ERA5 to understand the weather variabilities during the extreme weather conditions over the study area.

In detail, RegCM-SVN underestimates observed T2m ($UV_{10}$) on average by 0.8 °C (0.5 m/s) over the five studied stations together. Surface wind direction at Ras El Tin and Abu Qir weather stations showed the same NW predominant direction based on RegCM-SVN simulations, ERA5, and observed data. Over Port Said and El Arish, the RegCM-SVN results and ERA5 showed a predominant NNW direction, while observed data showed a predominant N direction. Over Cairo airport, both RegCM-SV and ERA5 showed a dominant wind direction ranging between N and NNE. In the same context, the observed data showed that the N direction is the predominant one. This may indicate that RegCM-SVN can simulate the studied parameters reasonably over SEL—providing the information necessary to understand the current weather and predicting future forecasts.

The RegCM-SVN simulations for the studied parameters showed a better agreement with ERA5 over the oceanic area than RegCM-SVN simulations over mountain regions, partly owing to the grid size used. To overcome this problem of less accurate weather simulations over the mountain region, a high-resolution simulation (1 km grid) should be used, as stated by Yasunaga et al. [39] and Goyette et al. [40]. These types of simulations are too costly to operate in regional climate modeling (dynamical downscaling) because they require a long time for simulations as well as huge data storage capacity. Thus, the authors agree with Hanssen-Bauer et al. [46] that, over the mountain regions, it is reasonable to use statistical downscaling tools.

The authors suggest that (1) adding more parameters to the physics of Albedo and advection scheme calculation; (2) improving the boundary condition using some observed stations side by side with ERA5; and (3) using a higher resolution of global land cover characteristics (up to 90 m) will eliminate the bias and increase the correlation between RegCM-SVN and observation. Improving the albedo parameterization for each land cover type will improve the RegCM-SVN results through more accurate calculations of atmospheric thermodynamics and regional radiation balance (Usha et al. [47]). Furthermore, more accurate calculation of advection scheme regarding regional stability issues has a positive impact on climate simulations (Park and Bretherton [48]). The authors' second suggestion follows [49], who stated that the impact of boundary conditions on the climate simulation is strong. Our third suggestion agrees with [50], who stated that the regional surface characteristics are very important in climate simulations and must be properly represented in models.

The annual T2m trend (from 1979 to 2018) over the SEL basin (0.4 °C/decade) showed lower estimation than that (0.5 °C/decade) estimated by Shaltout et al. [16], partly owing to the long time span of this study, which provided high simulation accuracy using the adopted model. In general, the current study stated that the SEL area described a significant warming trend during the study period that partly conformed to the previous findings of Domroes and El-Tantawi [51], Shaltout et al. [16], and Tonbol et al. [52]. On the other hand, the present simulated wind along the Egyptian Mediterranean coast has predominant direction NW and N directions, supporting the previous findings of Hamed [53], Hamed [54], Meligy [55], Elsharkawy et al. [56], and Mahfouz et al. [57]. Furthermore, the annual mean wind speed (from 1979 to 2018) was in close distribution with Mortensen et al. [58], partly confirming the quality of the current simulation.

The current study confirms that the annual trend of T2m has a significant spatial distribution (0.3–0.8 °C/decade) and the annual trend of UV$_{10}$ also has a significant spatial distribution (−0.15–0.05 m/s/decade). This finding may stimulate a future discussion about the effect of urbanization and the improvement RegCM-SVN configuration for land use conditions that are changing with time.

On the studied 40-year record over SEL, winter of 2010 and winter of 1983 were the warmest and coldest winters, respectively. In the same context, summer of 2015 and summer of 1984 were the warmest and coldest summers, respectively. These findings were consistent with the previous findings of Shaltout et al. [16], confirming the quality of the obtained simulations.

During 1979–2018, the annual variability in surface wind speed was markedly noticed than in T2m, especially during the winter season, owing to the frequent occurrence of atmospheric depressions (when N winds blow during winter), which largely affect the surface wind speed compared with T2m. The annual variability reached its maximum (minimum) values during winter (spring) for both surface wind speed and T2m as a result of the effect of winter atmospheric depressions.

During the study period, the seasonal features for T2m showed a general warming trend during the four seasons and showed a decreasing trend rate for wind speed during spring and summer.

There was a marked offset of seasonality timing over the SEL basin (north part of the study area) versus the southern part of the study area, partly indicating that seasons come earlier in the southern part than in the northern part. This finding can be explained by the fact that the surface air over the sea water takes a longer time to reach its maximum temperature than over the land area.

Finally, the SEL area is exposed to climate change, and its responses in terms of warming and non-significant wind speed trends become clearer than before. Moreover, the current study improves our understanding of the contribution of climate change to extreme weather over SEL. Thus, the decision-makers and climate scientists need to work together to review the Egyptian climate policy and work to improve it in order to discover innovative ways to turn the climatic challenges into suitably socio-economic opportunities.

The authors freely publish the RegCM-SVN code (name list) together with all the simulated data to be available for the scientific public via the MDPI Supplementary Material section.

**Supplementary Materials:** The following are available online at https://www.mdpi.com/article/10.3390/cli9100150/s1, Text document (Eastern_Levantine.text) describes the RegCM-SVN code (name list). Text document (RegCM-SVN_simulated_data.txt) provides the full link for the RegCM-SVN simulated data.

**Author Contributions:** Data curation, M.E. and M.S.; Formal analysis, K.T. and M.S.; Investigation, M.E., K.T. and M.S.; Methodology, M.E. and S.M.A.; Project administration, M.S.; Supervision, S.M.A. and M.S.; Writing—original draft, M.E. and M.S.; Writing—review & editing, M.S. and S.M.A. All authors have read and agreed to the published version of the manuscript.

**Funding:** This research received no external funding.

**Data Availability Statement:** Please refer to suggested Data Availability Statements in section "MDPI Research Data Policies" at https://www.mdpi.com/ethics (accessed on 4 October 2021).

**Acknowledgments:** This work was done as a part of the first author, Mohamed Elbessa's, dissertation. The authors would like to thank the ERASMUS+-FISHAQU project (No. 610071-EPP-1-2019-1-PTEPPKA2-CBHE-JP) for offering a highly reliable workstation to run the present simulations. This manuscript also benefited from comments provided by three anonymous reviewers, as well as the Special Issue editors. It is worth mentioning that the publication of this article was not possible without the help and use of Windographer software professional edition version 5. Thus, the authors specifically would like to thank the makers of Windographer software (Tom Lambert, Windographer Project Manager).

**Conflicts of Interest:** The authors declare no conflict of interest.

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
