# Peer review of "Dynamical Downscaling of Surface Air Temperature and Wind Field Variabilities over the Southeastern Levantine Basin, Mediterranean Sea"

_climate, doi:10.3390/cli9100150_

Round 1
Reviewer 1 Report
This is a nice application of the RegCM model to dynamically downscale ERA5 reanalysis with a focus on south-east Levant (coast of Egypt). The methodology and interpretation are sound, for example the correlations are derived with and without the seasonal cycle. Also the observational data from the five stations act as an independent reality measure for the evaluation of both ERA5 and model output, which is useful. Another noteworthy element of the study is the incorporation of the wind speed and direction in the model evaluation, a meteorοlogical variable that is often neglected.
line 40: and in many other studies, for example Let et al. (2020); Ehsan et al. (2020)
line 71: here are a few: Abdelwares et al. (2020); Osman et al. (2021)
lines 97-125: remove Italic formatting
lines 212-225: remove Bold formatting
line 212: remove "is also", is used twice
line 249: what does the "SVN" stands for in " RegCM-SVN"?
line 266: you mean "large-scale precipitation" in "5) scale precipitation"
pages 10-11 Figure 3: replace in the image title "corelation" with "correlation"
line 350: replace "described a" with "includes"
page 23 Figure 8 caption: which dataset is used here? model, ERA5 or station?
page 24 Figure 9 caption: which dataset is used here? model, ERA5 or station?
page 26 Figure 11&12 caption: which dataset is used here? model, ERA5 or station?
REFERENCES
Abdelwares M, Lelieveld J, Hadjinicolaou P, Zittis G, Wagdy A, Haggag M. Evaluation of A Regional Climate Model for the Eastern Nile Basin: Terrestrial and Atmospheric Water Balance. Atmosphere. 2019; 10(12):736. https://doi.org/10.3390/atmos10120736
Ehsan, M. A., Nicolì, D., Kucharski, F., Almazroui, M., Tippett, M. K., Bellucci, A., Ruggieri, P., & Kang, I.-S. (2020). Atlantic Ocean influence on Middle East summer surface air temperature. Npj Climate and Atmospheric Science, 3(1), 1–8. https://doi.org/10.1038/s41612-020-0109-1
Le, J.A., El-Askary, H.M., Allali, M. et al. Characterizing El Niño-Southern Oscillation Effects on the Blue Nile Yield and the Nile River Basin Precipitation using Empirical Mode Decomposition. Earth Syst Environ 4, 699–711 (2020). https://doi.org/10.1007/s41748-020-00192-4
Osman, M., Zittis, G., Haggag, M. et al. Optimizing Regional Climate Model Output for Hydro-Climate Applications in the Eastern Nile Basin. Earth Syst Environ 5, 185–200 (2021). https://doi.org/10.1007/s41748-021-00222-9
Author Response
Dear Reviewer 1
Thank you for your valuable comments which make our manuscript more obvious and clear

Reviewer 2 Report
Overall I found the topic interesting for the Journal's audience. However, the quality of presentation must be improved, in particular as regards the Introduction section (Section 1), where the scientific question addressed in the paper is not clearly posed, and the Data and Methods of Analysis section (Section 3), which should have an internal consistency with the Results Section (Section 4). The Discussions Section is missing (or should the reader look at the Summary And Conclusion Section?) and it is missing a general discussion of the findings in light of previous works. I don't see any particular need for Section 2 ("Climatological Background"), since it is not discussed in the subsequent text.
I encourage the authors to rewrite the manuscript in a more rigorous way.
The Introduction Section should be rewritten by clearly describing the general problem and the research gap that motivated the work. For example: "recently released global analyses represent a step-change tool for climate investigations, however their resolution is still inadequate to describe regional/local climates and a downscaling procedure is needed to provide more reliable data to policy makers." Furthermore a complete list of previous works should be reported stressing any research gaps and highlighting the innovative aspects of the present work.
The Data and Methods of Analysis Section should describe what is going to be shown in the Results Section. It is needed an internal consistency between the two sections: what is described in the former section, is shown in the latter and viceversa. For example the Fourier analysis is not described, but Figures 10a-b show the results based on this method.
The results shown in Section 4 are not discussed in light of previous works and with respect to the scientific question posed in the Introduction and my feeling is that a Discussion section would help the reader to understand the significance of the contents shown in the manuscript, along with the innovative aspects.
Finally, although I'm unfit to evaluate the quality of the English language, my feeling is that a language review is needed before publication. Furthermore the author's guidelines should be followed more strictly.
I'm attaching an amended version of the manuscript with some additional major/minor comments (double click for pop-up note).

Author Response
Dear Reviewer 2
Thank you for your valuable comments which make our manuscript more obvious and clear

Reviewer 3 Report
The Review is in the attached PDF file.

Author Response
Dear Reviewer 3
Thank you for your valuable comments which make our manuscript more obvious and clear

Round 2
Reviewer 2 Report
I found the re-submitted version of the manuscript a large improvement over the initial submission and commend the authors for taking the time to thoroughly review feedback and make changes to the text. I believe the manuscript is more suitable for publication. However:
(i) the authors did not provide any reason for adding the section "Climatological Background". I have the feeling that this section is not necessary, unless it is explicitly commented/discussed with the findings shown in the manuscript, which is not in my opinion.
(ii) I encourage the authors to pay more attention to the text editing (i.e., unnecessary capitalized words, nested parentheses, etc...).

Author Response

(The authors gave the same response as above.)

Reviewer 3 Report
Review on manuscript MDPI climate-1358098 (Round-2):
“Dynamical downscaling of surface air temperature and wind field variabilities over the South-Eastern Levantine Basin, Mediterranean Sea” by authors: Mohamed El Bessa, Saad Mesbah Abdelrahman, Kareem Tonbol, and Mohamed Shaltout Caravan
Comments
The authors took into account the remarks made by the Reviewer.
However, in the revised version of the manuscript, there are typos, look like round brackets “()” in line 214.
There is only one point that I did not understand. It was the response to Comment 36:
“We plan to publish all data parameters (Alexandria University website and the MDPI Climate journal supports the Complementary Material) after the acceptance of the paper” “
At the same time, in the answers below, the authors wrote: “No Supplementary Materials”.
Further in the revised version of the manuscript (line 791 – 793) authors wrote the next:
“The authors plan to freely publish the RegCM-SVN code (name list) together with all the simulated data to be available for the scientific public via the Alexandria University website (Corresponding author home university) and in the MDPI supplementary material section.”
If the authors decide to use the Supplementary Material option, they must submit the material in the Supplementary section during the review process, i.e., before accepting the manuscript. Also, the authors have to add in the manuscript a text that looks like the following MDPI Template:
“Supplementary Materials: The following are available online at www.mdpi.com/xxx/s1, Figure S1: title, Table S1: title, Video S1: title.”

Author Response

(The authors gave the same response as above.)
